# Remittances, Migration and Gross Domestic Product from Romania's Perspective

**Anca Mehedintu [1], Georgeta Soava [1],\* and Mihaela Sterpu [2]**

[1]   Department of Statistics and Economic Informatics, University of Craiova, A.I. Cuza 13,
     200585 Craiova, Romania; ancamehedintu@yahoo.com
[2]   Department of Mathematics, University of Craiova, A.I. Cuza 13, 200585 Craiova, Romania;
     msterpu@inf.ucv.ro
\*    Correspondence: georgetasoava@yahoo.ro; Tel.: +40-722-985-087

**Abstract:** This study analyzes the evolution and trends of the share of remittances in gross domestic product (GDP) and the influence of migration on remittances in Romania. The analysis on data from Eurostat over 2008–2017 has three components: a statistical analysis, an estimation of evolution of indicators, and an estimation of impact of migration on remittances, using polynomial-time regression and difference equation models, respectively. The results showed that GDP and GDP/capita had a permanent increase, meaning an improvement in the standard of living in Romania, while the other indicators had an evolution with a period of sharp decline triggered by the global crisis, followed by a slow growth. We may conclude that the remittances represented and still represent a relatively stable financial resource for Romania as for the other emerging countries in Europe, affecting in a positive way the standard of living of the citizens, although their value has a tendency to decrease. At the same time, the negative effects of remittances, dependence on money received from migrants and the exodus of "brains" and skilled workers, must be considered, implying the necessity of government policies for a better use of remittances, i.e., mainly for investments and less for consumption.

**Keywords:** remittances; migration; regression models; GDP; Romania

## 1. Introduction

At the beginning of the new millennium, more and more international development agencies and governments are considering the potential of migration and remittances to stimulate development in developing countries. Globalization makes its presence even more pronounced, especially for the countries of Eastern Europe, by increasing the labor force migration after 1989, strongly influencing the economy of each state. The number of emigrants from these countries and the influx of remittances have increased significantly from year to year, yet, while residing abroad, the emigrants remain connected with their native countries, thus diminishing the loss of their identity and their separation from the countries of origin.

After the political transformations of Central and Southeast Europe (Albania, Bosnia and Herzegovina, Bulgaria, Croatia, Czech Republic, Greece, Hungary, Poland, Romania, Serbia, Slovakia, Slovenia, Macedonia, and Ukraine) in the early 1990s followed by a long period of time (over a quarter of a century), exhibiting large and persistent migration flows from east to west, dominated by young people in general and by young people with higher education, the countries of southeastern Europe experienced massive labor outflows that lasted until the end of 2012 as a consequence of their accession to the EU (in 2004: Estonia, Hungary, Latvia, Lithuania, Poland, Czech Republic, Slovakia, and Slovenia; in 2007, Romania and Bulgaria; and in 2013, Croatia), when citizens were free to travel and work. The determinants of migration were the differences between the level of income per capita, the quality

of government policies, and the employment prospects. Thus, the countries with the highest gross domestic product (GDP) per capita attracted the most migrants, the main destinations of emigrants being Western European countries (8 out of 10 migrants), mainly Germany, Italy, and Spain (60%) and the United States (about 1 in 10 emigrants or 9%) [1].

Since the 1990s, the borders of the states and the issues related to migration have been found among the European problems of major interest [2]. There has been an increase in permanent migration (following the development of information and communication technology, health, and education, sectors that require high skilled labor force) and temporary migration (the demand for unskilled foreign labor has increased, especially in agriculture, construction, and public works, as well as domestic services—the cases of Spain, Italy, Portugal, and Greece).

The number of migrants in the developed countries of the European Union has increased considerably, with tens of thousands of people leaving the country of residence for a greater gain and thus a better life. Thus, the citizens of emerging countries will continue to migrate to developed countries as long as, in these countries, there will be a demand for labor and a stronger economic development than in the countries from which they come. However, it should be noted that not all migrants are immediately successful in finding a job and can start sending money to their country of origin.

The increase in migration since the 1990s as well as the increasing importance of remittances as a source of financing for development determines the decision makers to consider how best to use these human and financial resources. In this regard, a number of researchers have turned their attention to the topic of migration and how migrants contribute to the economic development of their countries of origin and to the impact of highly skilled migrants on the countries where they work. The demand for highly skilled workers can be met to a great extent by developing countries, with the direct benefits of "brain migration" still highly appreciated. The import of specialists still takes place, even if its significance is lower. An increase in the reverse flow of specialists is expected from the rich to the least developed countries as a result of the reduction of the demand for highly qualified personnel due to the increase of the economic efficiency in the developed countries. At the same time, direct capital and investments will go to poor countries, attracting specialists from rich countries [3].

In this context, Romania is an important case study, being among the first markets to receive remittances in Europe for almost 10 years, with a volume of remittances per year of over 6 billion Euro. According to the World Bank statistics [4], in 2015, almost 5 million Romanians worked abroad and 68% of these sent money to their families (about 3.4 million Romanians). Italy, with over 1 million Romanian migrants, is the main country from which Romanians send money, followed by Spain with over 650 thousand Romanian migrants and Germany with 590 thousand Romanian migrants.

The free movement and the opening of the labor market (globalization and the need to cover the demographic deficit in the developed countries with an aging population) stimulated the mobility of the labor force, thus getting Romania to have in 2017 a number of emigrants representing 15% of Romania's population [5].

The increase in the number of migrant workers has led to an increase in remittances in the countries of origin and thus their share in the GDP, thus being an important source of external financial flows, which can produce significant changes both at the macroeconomic level and at the household level. The EU countries that rank first in the share of GDP remittances are Latvia and Croatia, with over 4% [6].

In Romania, the percentage of remittances in GDP is around 1.86%, the results could be significant if the remittances would be analyzed in relation to the underground economy, which employs over 1.2 million Romanians and holds over 22% of Romania's GDP at the level of 2017 [7].

According to the World Bank report in 2018 [8], after three years of consecutive decline, remittances sent in 2017 to the countries of central and southeastern Europe increased by 20.9%. In this context, we note that, in 2018, Romania is placed the first among the EU countries that received the largest amounts

of money from Romanians abroad, 4117 billion euros, up from 2017 (3795 billion euros), exceeding Poland in terms of the amounts sent home by the citizens of the diaspora [9].

According to the European Strategy [10], EU member states must aim to create a more sustainable economic future. In line with the objectives of the Europe 2020 Strategy, the European Employment Strategy aims to create more and better jobs throughout the European Union. Actions are needed to support a structural adjustment that will lead to more and better jobs. In the last years, the active population in the EU28 has registered a continuous reduction, explained by the aging of the population and the decrease of the number of the active persons and, at the same time, the decrease of the employment rate determined by the restructuring of many activities under the influence of the market demand and of the progress technical [5]. Given the current trends, the decline started in 2011 will continue until 2030 as the age of the active population in the European Union will lead to a decrease in the number of people employed up to about 20 million people [2].

Romania is a member of the European Union that occupies one of the first places in terms of the number of emigrants in the developed countries of the EU and of the amount of remittances received from emigrants. This study has as the main object an analysis of the evolution and trends of remittances, migration, nominal and per capita gross domestic product, and the relationships between them for Romania over the period 2008–2017.

Thus, Section 2 presents an extended review of literature on empirical studies related to this topics, demonstrating the importance of such a study.

The analytical study in Section 3 contains three components. Firstly, in order to create an overview of how these indicators evolved during the period of interest, a statistical analysis of the indicators is performed. The relationship between remittances and gross domestic product is represented by means of a new indicator, namely share of remittances into GDP, obtained by dividing the annual value of remittances to the corresponding nominal GDP. Secondly, an estimation of evolution and trends for all the indicators (remittances, migration, nominal and per capita gross domestic product, and share of remittances in GDP) is realized using three polynomial-time regression models. Finally, the relationship between remittances and migration is estimated by means of two first order difference equations.

The empirical results are interpreted economically in the last section, that contains conclusions of the analytical study and provides possible directions for Romanian government or European policies related to the addressed issues.

The results showed that GDP and GDP/capita have seen a permanent increase, meaning that, in Romania, there has been an improvement in the standard of living, while the other indicators have had an evolution with a sharp decline triggered by the global financial crisis, followed by a slow growth.

We may conclude that the remittances represented and still represent a relatively stable financial resource for Romania as for the other emerging countries in Europe, affecting in a positive way the standard of living of the citizens, although their value has a tendency to decrease. At the same time, the negative effects of remittances and the possible risks due to the reduction of the employment intention are the dependence on the amounts of money received from outside the borders of the country and the exodus of "brains" and skilled workers. As a consequence, the necessity of government policies for a better use of remittances are imperative.

The main contribution and novelty of this study is the type of analysis (using linear, quadratic and cubic regression models, and difference models) chosen for this configuration of indicators.

Similar estimations could be performed for other developing countries from Europe.

## 2. Literature Review

The debate on migration and development is not new, but with the changes that have taken place worldwide, there has been a growing interest in revising empirical research to achieve a more realistic picture. Starting with the year 2000, the study of the relationship between migration and development was initially focused on its consequences within the society of origin of the migrants viewed through the decision makers and scientists.

Migration is a reality of modern society, with economic, social, and political implications, and in this sense, some EU members fear that they could be overcome by this phenomenon [11]. Migration is considered a factor for stimulating global markets and as a tool for regulating imbalances in regional and local labor markets. Emigration represents the total number of long-term emigrants considered in a reference year [12]. Migration represents "the movement of people across international borders and has major implications for economic development and poverty reduction, both in the countries of origin and in the designation of migrants" [4].

Migration has profound economic consequences for the countries of origin, some with positive implications, others with worrying consequences. One of the main positive effects of international migration concerns the financial transfers to the country of origin of the senders, which are often seen as offsetting the "brain drain" and the flow of human capital. In many developing countries, migration is aimed at improving both the standard of living of the emigrant and those of the family left in the country of origin through remittances that usually far outweigh the initial expenses or income that could have been earned in the country of origin. Remittances increase the country's income from external sources, and as a result, it increases not only the standard of living of the recipients but also the level of local economic development, through consumption and investment. Despite the positive aspects of financial rewards, separation from a family can involve high emotional costs for both those leaving for work and those remaining in the country of origin.

According to EU legislation, citizens of EU states can travel freely within EU borders; on 1 January 2018, the number of third-country nationals and residence in an EU member state was 22.3 million (representing 4.4% of the population of EU-28 states) and 17.6 million people living in one of the EU member states were citizens of another EU member state. Regarding the country of origin, 38.2 million people born outside the EU-28 lived in one EU State, while 21.8 million people were born in another EU member state than the one in which they resided (for example, only in Ireland, Cyprus, Luxembourg, Hungary, Malta, and Slovakia, the number of persons born in other EU State was higher than the number of persons born outside the EU-28). The largest number of migrants living in an EU State are in Germany (9.7 million people), the United Kingdom (6.3 million), Italy (5.1 million), France (4.7 million), and Spain (4.6 million). The main groups of EU citizens living in other EU member states are Romanians, Polish, Italians, Portuguese, and British. The highest share of the foreign population was registered in Luxembourg, and the lowest was in Romania. At the beginning of 2017, 7.5% of the total population of the 28 EU member states lived outside the economies of origin [12].

EU migration policies are increasingly aiming to attract a certain profile of migrants, most often to remedy certain skills shortages. Migrant beneficiary countries have increased the potential for migration development by creating legal channels for migration and integration policies that favored the socioeconomic mobility of migrants and avoided their marginalization [13].

Many researchers from European Union countries have studied the major processes involved in labor migration, focusing in particular on the changes that have come with joining the European Union [14]. The accession of the eastern European states to the EU was followed by an increase in the migration process from these countries to the already member states, which provides the foundations of significant flows of remittances, leading to a macroeconomic impact on both the receiving and the migrant origin countries [15,16]. At the same time, there was an opening of the labor market from three EU countries—Ireland, Sweden, and the United Kingdom—and Norway outside the EU for migrants from the new members, while the others opted for a transitional period and/or imposed conditions for migrants.

Due to demographic and economic imbalances, migration flows are attracted by the fastest growing economies. Thus, in the short term, Europe needs to increase labor mobility between EU states, given that, in some regions, unemployment is extremely high, while others may face skills shortages. In the long term, however, it will not be enough to reduce the gaps in European labor markets, especially since there are still many Europeans who are not prepared to accept even more migrants [17].

Remittances are amounts of money transferred from the host country by migrant workers to persons in the country of origin, who are in their care in their home countries. "Remittances" are current private transfers made by nonresidents to their country of origin by workers residing in the host country for more than one year [4].

Remittances are more than money transfers; they represent in fact links between migrants and their families in the country of origin [18], being considered the largest source of cash in the world [19]. Remittances are the result of labor mobility and occur due to the migration of categories of people from countries with less developed economies to developed countries [20].

Remittances consist of both transfer items (transfers sent by migrants to home economies) and income (personal transfers and employee compensation) [21]. Remittances and their impact on economic development are worth studying, both at the macroeconomic level and at the household level. The beneficiaries of remittances are, at the microeconomic level, for the families remaining in the country [22,23] and, at the macroeconomic level, the effects of remittances that are manifested on inflation, exports and imports by increasing domestic consumption and supporting the balance of payments [24]. The activity carried out by immigrants has positive effects for both the receiving state (as a result of the incomes and production achieved) and for the country of origin (through remittances and repatriation of the know-how accumulated by the migrant labor force in the host countries). Thus, it can be said that an increase in the volume of migration can increase the remittances that have a special implication on the development of the countries of origin and, at the same time, the immigrant labor force can support the economic activities where the force of domestic work cannot cover them, either due to lack of qualification or lack of interest for those sectors.

The remittance flow is an important source of funds for developing countries, as it is the main source of currency and can influence the balance of payments [25], contributing significantly to the development of the national economy [26,27]. Increasing foreign currency reserves means that the supply of foreign currency exchange increases and, in turn, will affect the exchange rate [28].

In this context, the emigration of the labor force has strongly affected the economy of these countries, having both negative and positive effects on the economic growth of the eastern European states. As a positive effect, for the country of origin, we notice a reduction in unemployment [29], increasing the individual welfare of migrants and their families, which also leads to an economic development of the country of origin through the money sent to the country [30,31]. At the same time, the European Union has had considerable advantages due to the cross-border mobility of the labor force and, in particular, of the highly qualified workers with higher education. Thus, it can be said that long-term positive effects can be recorded for both beneficiary countries, both the originating ones (as remittances can help stimulate investments) and destination country [32].

The negative effects are manifested in the countries of origin by the fact that the departure of a significant part of the skilled workforce coincided with the aging of the population in many countries of eastern Europe, which had major effects on their production and productivity, leading to reduced competitiveness [33–36]. Emigration significantly marked the demographics of the population of the countries of origin, which recorded a stagnant or declining population.

The increase in remittances can be supported not by the economies of the migrants sent during the difficult economic periods in the countries of origin but by the numerical increase of the migrants as a result of the shocks on the incomes [37]. Most studies on remittances of workers analyzed the effects of remittance entries, which were influenced by changes in the macroeconomic conditions of the host countries [38] more than by the changes in the country of origin [39]. The flow of remittances out of a country depends on the plans of the migrants regarding their return to the host country; if they stay temporarily, they send more money than they intend to stay for a long time.

Remittances have a positive impact on economic growth [40–45] and on the credit rating of a country, providing an important and stable source of foreign currency that can reduce panic for investors; it can cope with the crisis of balance of payments and can be used for development projects [46].

At the same time, remittances contribute to a better allocation of resources in the country of origin, thus stimulating the aggregate demand for goods and services by increasing the productivity generated by consumption and investment [47] and increasing the income and productivity by reducing the unemployment rate due to the mobility of the unemployed [29].

Some researchers believe that remittance flow has a negative effect on economic growth [48,49]. Thus, remittances are a factor that stimulates the imports of foreign substitution products into the internal market [50], and on the other hand, the consumption of imported products is higher than the "domestic consumption" of similar products in these countries [51].

Other researchers consider that, depending on certain periods, the remittances can have positive or negative effects [52,53] or no effects [25,54].

The analysis of the long-term and short-term impact of remittances on financial development in developing countries demonstrated the existence of a long-term positive and slightly positive short-term relationship, with the exception of countries with very low incomes [55]. Remittance flows to developing countries were more stable than other financial flows even when the global economy was affected by the global financial crisis of 2009 [56,57]. Remittances can act as a shock transmitter for beneficiary countries during the economic decline of migrant host countries, but they can also act as a buffer in stabilizing production volatility and consumption caused by internal negative shocks, such as natural disasters [58,59].

The correlations between remittances, population migration, poverty, and economic growth from the macro- and microeconomic perspective were analyzed, and the authors consider that remittances increase with increasing emigration and have a positive effect on poverty reduction but that the impact of remittances on economic growth is difficult to quantify [60].

In the short term, remittances caused by population migration lead to poverty reduction [61,62]; in the long term, remittances can stimulate growth but the effect is only significant at low levels of financial development [63].

How remittances affect household well-being, however, depends on the countries where the labor force migrates. Consumption of durable goods, health, and housing has been found to be significantly higher in households receiving remittances [64] and financial development and remittance flows lead to an improvement in poverty in developing countries [65]. Some studies conducted in Central Asian countries show that both international migration and remittances significantly reduce poverty [66].

Analysis of the influence of size and effects of remittances and emigration on poverty on the case of 4 western Balkan countries Macedonia, Albania, Serbia, and Kosovo show that, while remittances will increase, emigration will decrease slightly, but poverty levels may be reduced by the contribution of remittances [67].

The trends regarding remittances of migrants and their impact on economic growth have been widely analyzed in different studies. The relative importance of remittances of migrant workers as a source of financing for economic growth, investment, and distribution of income in beneficiary economies was emphasized [24], with international migration being considered a precondition for remittances [68].

In recent years, there has been a stagnation and even a decrease in the flow of remittances to the countries of residence due to the economic growth at the level of the European Union (EU) and the strengthening of the euro. However, long-term risks remain; in many countries that are sources of remittances, anti-immigration sentiment is growing and immigration policies are becoming more stringent. Between 2008 and 2014, remittance outflows decreased from EUR 13.9 billion in 2008 to EUR 1.4 billion in 2014. The net inflows of incomes generated by EU citizens through their work abroad in 2017 compared to 2015 decreased by EUR 17.0 billion, and the net flows of migrants' personal transfers to their home economies increased dynamically from EUR 19.0 billion in 2013 to EUR 22.0 billion in 2017 [12].

The countries of destination of the Romanian migrants changed according to the migratory regime in which the international mobility took place, between 1990–1995, when the entry into various

countries of Western Europe was severely limited; the main destination country was Germany (because the German ethnic groups in Romania used the preferential immigration institution on the basis of ethnicity to relocate definitively). An important segment of the population of Transylvania turned to Hungary and, in order to carry out the small commercial traffic or for seasonal work the Romanian citizens, turned to Israel, Turkey, the former Yugoslavia, and Greece [69]. Between 1996 and 2002, migration to western European countries was extended, with migrants choosing Italy and Spain. The next phase of labor migration was marked by the elimination of visas for countries in the Schengen area for Romanian citizens (1 January 2002), and covered the period preceding the economic-financial crisis, in which labor migration from Romania was a widespread phenomenon, with a reorientation of the Romanian migratory flows, Germany remaining one of the countries preferred by the Romanians, but due to the strictest control of the emigration, the Romanians mainly oriented towards 2 countries of destination: Italy (the highest number of migrants among all the migrants was registered) and Spain, that offered them a series of opportunities for increasing the standard of living [70]. There is a spectacular increase in the number of emigrants of Romanian origin after 2002; thus, in 1990, approximately 286,800 Romanian migrants were officially registered, and in a decade, their number increased by 63%, reaching 469,300 people, while in the next decade, the growth would be even more remarkable, reaching 2,769,053 immigrants of Romanian origin. It is also estimated that the process of emigration from Romania will continue in the next period [69].

Labor migration is a phenomenon that can produce a very complex set of economic effects in the countries of origin. Remittances are the most visible result of migration, representing the second largest source of external financing after foreign direct investment (FDI) in the economy. It also covers a deficit of incomes for household consumption: access to services, health, and education. The causes of the migration from Romania were mostly economic in nature (the concrete needs of the household and the economic problems at the Romanian level) and the need to try something new by imitating the success models of other migrants in the community.

In some cases, the emigration of some members of the households means that, due to receiving remittances, the income generated by their activities in the country of origin can be reduced. However, the remaining members in the country can make various investments, which can lead to the economic growth of the respective community. Remittances have been a relatively stable source for Romania, but their excessive dependence on growth and development is not without danger. Thus, the value of remittances tends to decrease once the migrant community becomes more stable in the country of destination. At the household level, the dependence on remittances leads to the neglect of the productive activities of the family, and at the community and regional levels, it leads not only to an increase of inequality between the families receiving remittances and those that do not receive remittances but also to the increase of inflation. At the same time, there was a marked decrease of them also during the financial crisis due to the deterioration of the economic conditions in the host countries.

Remittances in Romania increased significantly from 0.9 billion euros in 2000 to 1.03 billion in 2001 to 5.5 billion euros in 2006 and reached 6.65 billion in 2009. However, the highest value was in 2008 at EUR 8.64 billion. Optimal management of labor migration could stimulate social development at the personal, family, and community levels, leading to a reduced rate of unemployment and poverty. In 2009, there was a drop in remittances due to the financial crisis, a decrease that continued in the following years.

We may conclude that migration affects both the standard of living of the citizens who arrive in the developed countries and the countries where they arrive. Thus, it is the responsibility of the governments of both developed and developing countries to find ways of maximizing benefits and minimizing the disadvantages of migration and remittance flows. Remittances that enter the country of origin lead to an increase in incomes from external sources, with direct effects in increasing the standard of living of those who receive them and of local economic development (by increasing consumption and investments).

Possible negative effects of remittances and possible risks consist of increasing inequalities at the community level, decreasing the intention to engage in productive activities on the national market, reliance on remittances, and the emergence of inflationary pressure because the excessive demand for land and houses leads to artificial increase of their prices and, not least, the existence of the brain drain phenomenon and the migration of skilled workers. In the period after 1990, Romania experienced a general destabilization of the economy, which led to a significant restructuring of the economic field, which materialized in the disappearance of large companies and the reorganization of others, which produced a great financial problem for the citizens. This has led to mass migration to countries such as Italy or Spain, and with the elimination of visas for entry into the EU, labor migration has become one of the most important phenomena. Analyzing the value of remittances entering the country, Romania is in the top 10 at the global and European levels [71].

Starting from the importance of remittances, states need to show increased interest in finding the most appropriate ways to use them efficiently, especially since for Romania as for many other developing countries, remittances are one of the main sources of external financing (represents a significant percentage of GDP).

## 3. Models, Empirical Estimations, and Results

The macroeconomic indicators used for the analysis are the number of emigrants from Romania (MG), the remittances received by Romania from the EU countries expressed in millions of euros (REM_EU) and from worldwide expressed in millions of euros (REM_W), the gross domestic product at nominal values (GDP—expressed in millions of euros) and per capita (GDP_C) expressed in euros, and the share of remittances from the EU level (S_REM_EU) and from the global level (S_REM_W) in the gross domestic product. The values of five of these indicators for the period 2008–2017 are taken from the Eurostat database [72]. The statistical data collected from Eurostat have no values for migration and remittances until 2008, which is why we chose the study period 2008–2017. The annual values of S_REM_EU and S_REM_W were obtained by dividing the nominal value of remittances by the nominal GDP.

The empirical study carried out in this paper has three components. First, the statistical analysis of the data is performed. Second, estimations of the evolution of the economic indicators mentioned above by means of 3 polynomial-time regression models are realized. The third part of the study aims to estimate the impact of migration on remittances using the following two linear difference equation models.

The theoretical models are estimated and the data are processed using the econometric, data processing, and analysis software EViews 9.5 [73]. The empirical results are then analyzed and compared.

### 3.1. Statistical Data Analysis

The first years after Romania's integration to the European Union was accompanied by a significant increase of the departures to work abroad, after which the pace of departure was slower; with the crisis, it slowed down and a redistribution of the existing migrants in the countries of the European Union has taken place. The main destination countries for Romanians are Canada (20%), Germany (19%), and USA (18%). After 2008, it is noted that the number of Romanian citizens left to work abroad decreased year by year until 2013 (Table 1), followed by a slight increase, but in 2017, compared to 2008, the number of migrants was lower 20%.

The smallest number of emigrants from Romania in the analyzed period was in 2013 (161,755 persons), after which there is again an increase to over 200,000 people per year (Figure 1).

The crisis was also felt by the Romanian workers from the EU countries; thus, they began to reorient themselves, migrating from the peripheral economies of the EU (Spain, Greece, or Italy) which attracted the bulk of the migrant flow towards more economically developed states, which offered other job opportunities (England, Norway, Finland, Sweden, and Denmark).

**Table 1.** Values of the analyzed indicators (MG, REM_EU, REM_W, GDP, GDP_C, S_REM_EU, and S_REM_W), period 2008–2017.

| Years | MG | REM_EU | REM_W | GDP | GDP_C | S_REM_W | S_REM_EU |
|---|---|---|---|---|---|---|---|
| 2008 | 302,796 | 4792.0 | 5156.4 | 146,590.6 | 7100 | 3.518 | 3.269 |
| 2009 | 246,626 | 2661.6 | 3020.8 | 125,213.9 | 6100 | 2.413 | 2.126 |
| 2010 | 197,985 | 2003.7 | 2448.8 | 125,408.8 | 6200 | 1.953 | 1.598 |
| 2011 | 195,551 | 1844.0 | 2295.5 | 131,925.4 | 6500 | 1.740 | 1.398 |
| 2012 | 170,186 | 1801.6 | 2286.7 | 133,147.1 | 6600 | 1.717 | 1.353 |
| 2013 | 161,755 | 1634.3 | 2098.4 | 143,801.6 | 7200 | 1.459 | 1.136 |
| 2014 | 172,871 | 1548.9 | 2001.7 | 150,458.0 | 7600 | 1.330 | 1.029 |
| 2015 | 194,718 | 1637.4 | 2176.0 | 160,297.8 | 8100 | 1.357 | 1.021 |
| 2016 | 207,578 | 2141.0 | 2449.0 | 170,393.6 | 8600 | 1.437 | 1.257 |
| 2017 | 242,193 | 2460.6 | 2822.8 | 187,516.8 | 9600 | 1.505 | 1.312 |

Note: MG—the number of emigrants from Romania; REM_EU—the remittances received by Romania from the EU countries; REM_W—the remittances received by Romania from worldwide; GDP—the gross domestic product at nominal values; GDP_C—the gross domestic product per capita; S_REM_EU—share of remittances from the EU level in the gross domestic product; S_REM_W—the share of remittances from the global level in the gross domestic product.

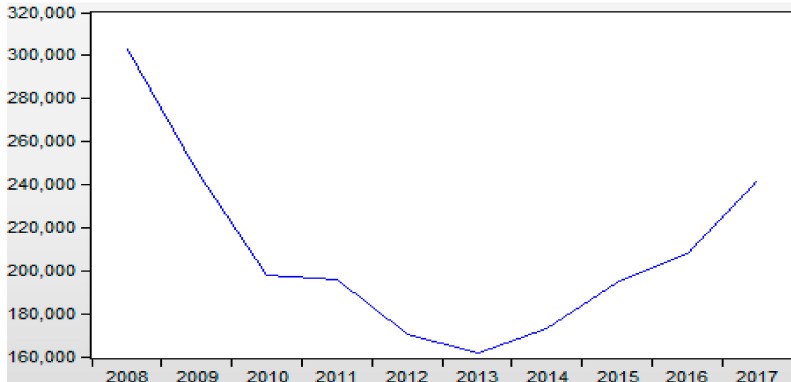

**Figure 1.** Evolution of the number of emigrants over the period 2008–2017.

The evolution of remittances presented in Table 2 shows that the value of the remittances had a tendency to increase until the global crisis started, after which the remittance level was strongly affected; thus, in 2009, compared to 2008, their value decreased by about 40%, both for remittances received from EU states (44.46%) and those received from all states (41.42%).

**Table 2.** Evolution of the indicators during the period 2008–2017 (%).

| | 2009/ 2008 | 2010/ 2009 | 2011/ 2010 | 2012/ 2011 | 2013/ 2012 | 2014/ 2013 | 2015/ 2014 | 2016/ 2015 | 2017/ 2016 | 2017/ 2008 |
|---|---|---|---|---|---|---|---|---|---|---|
| MG | −18.55 | −19.72 | −1.23 | −12.97 | −4.95 | 6.87 | 12.64 | 6.60 | 16.68 | −20.01 |
| REM_EU | −44.46 | −24.72 | −7.97 | −2.30 | −9.29 | −5.23 | 5.71 | 30.76 | 14.93 | −48.65 |
| REM_W | −41.42 | −18.94 | −6.26 | −0.38 | −8.23 | −4.61 | 8.71 | 12.55 | 15.26 | −45.26 |
| GDP | −14.58 | 0.16 | 5.20 | 0.93 | 8.00 | 4.63 | 6.54 | 6.30 | 10.05 | 27.92 |
| GDP_C | −14.08 | 1.64 | 4.84 | 1.54 | 9.09 | 5.56 | 6.58 | 6.17 | 11.63 | 35.21 |
| S_REM_W | −31.42 | −19.06 | −10.89 | −1.30 | −15.03 | −8.83 | 2.03 | 5.88 | 4.74 | −57.20 |
| S_REM_EU | −34.98 | −24.84 | −12.52 | −3.20 | −16.01 | −9.42 | −0.78 | 23.01 | 4.43 | −59.86 |

In 2017 compared to 2008, the value of remittances decreased by almost half, both those from EU (from 4792 million euros in 2008 to 2460.6 million euros in 2017 or 46.95%) and those received from worldwide (5156.4 million in 2008 to 2822.8 million in 2017, 45.26%). It is noted that, with the global crisis, the remittances decreased dramatically compared to 2008 (which is considered their top year) (Figure 2).

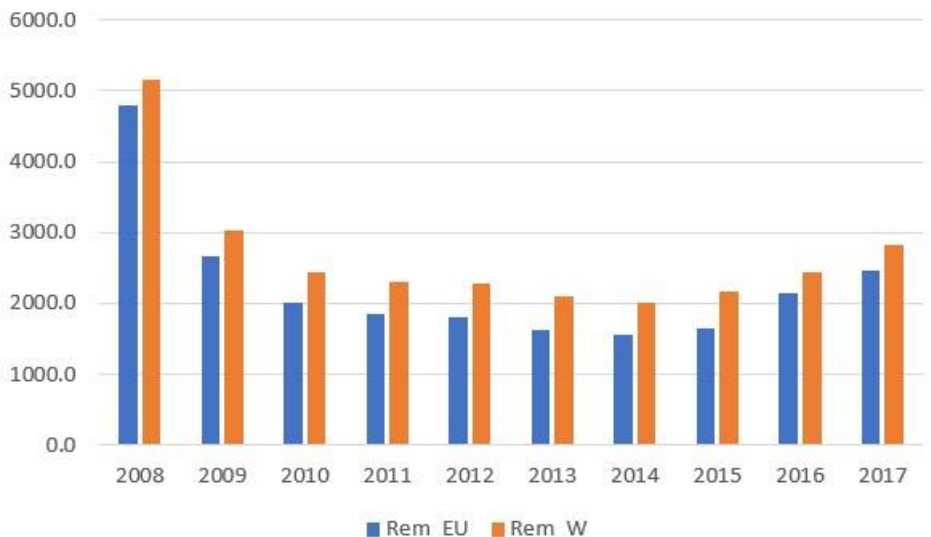

**Figure 2.** Evolution of remittances over the period 2008–2017.

Analyzing the evolution of GDP and GDP/capita, it is noted that both indicators showed a permanent increase except for 2009 compared to 2008 when they registered a consistent decrease due to the financial crisis. It is also noted that the GDP in nominal values increased with respect to the year 2008 by 27.92% and the GDP per capita increased by 35.21%, which means that there has been an improvement in the standard of living in Romania.

Regarding the share of remittances in GDP, there is a drastic decrease of around 60% in 2017 compared to 2008, both of those coming from the European countries (59.86%) and those coming from the global level (57.20%), as an effect of the global crisis; thus, the citizens sent less money to the country, but at the same time, their number decreased. After a continuous decline over 6 years, there is a slight recovery.

The descriptive analysis of the 7 indicators is summarized in Table 3.

**Table 3.** Descriptive analysis of the indicators, period 2008–2017.

| Indicator | Mean | Median | Max | Min | Std. Dev. | Skw | Kurt | Jarq.-Bera | Prob |
|---|---|---|---|---|---|---|---|---|---|
| MG | 209,226 | 196,768 | 302,796 | 161,755 | 43,290.7 | 0.99 | 3.13 | 1.64 | 0.440 |
| REM_EU | 2252.5 | 1923.85 | 4792 | 1548.9 | 963.77 | 2.02 | 6.04 | 10.68 | 0.005 |
| REM_W | 2675.6 | 2372.15 | 5156 | 2001.7 | 926.799 | 2.14 | 6.41 | 12.49 | 0.002 |
| GDP | 147,475 | 145,196 | 187,517 | 125,213.9 | 20,400.9 | 0.69 | 2.46 | 0.91 | 0.635 |
| GDP_C | 7360 | 7150 | 9600 | 6100 | 1130.59 | 0.72 | 2.50 | 0.97 | 0.615 |
| S_REM_W | 1.55 | 1.33 | 3.27 | 1.021474 | 0.6843 | 1.78 | 5.09 | 7.08 | 0.029 |
| S_REM_EU | 1.84 | 1.61 | 3.52 | 1.330404 | 0.6742 | 1.71 | 4.87 | 6.32 | 0.043 |

We note that three indicators (MG, REM_EU, and REM_W) had only in 3 years (2008, 2009, and 2017) values above average; GDP and GDP_C have values above average during the last 4 years studied, and regarding the weight of remittances in GDP, we find that, at the S_REM_W level, the values exceed the average in 5 years (2008–2012) while S_REM_EU has values exceeding the average only in 2008 and 2009, the rest being drastic decreases, the values having an exaggerated high dispersion. GDP and GDP_C have normal relative distribution.

The analysis of remittance values shows that Romania obtains an average of 84.19% of transfers from EU countries, while 15.81% comes from the rest of the world, which means that Romania's population is mainly oriented towards the EU for finding a job.

The distribution of all indicators has an asymmetry to the right, being almost mesokurtic, slightly sharp compared to the normal distribution for MG; leptokurtic for REM_EU, REM_W, S_REM_EU, and S_REM_W; and leptokurtic for GDP and GDP_C.

## 3.2. Estimation of Evolution by Means of Polynomial-Time Regression

The effects of migration and remittances have been increasingly studied in recent years, in which context for a series of micro- or macroeconomic approaches have been developed. Migration cannot be separated (remitted) from remittances because people working in another state create income that they send to the remaining family in the country, which can be used either for investment or for consumption, leading to economic growth.

Different regression models have been used frequently, on groups of states or on individual states, obtaining different results. For instance, using the Panel Smooth Transition Regression model on a sample of 49 developing states, remittances were found to have a positive impact on the level of economic development [45]. By applying the generalized moments method on a panel of 100 developing countries, remittances have been shown to contribute to GDP growth in the countries analyzed [31].

As Romania occupies a top position in terms of the number of emigrants and the value of received remittances, numerous studies have analyzed different aspects related to remittances. The macroeconomic determinants of remittances to Romania were analyzed using the panel data model with several variables with potential influence on remittances, proving that the traditional influence factors (distance, diaspora concentration, or unemployment rate) are, currently, less important than the difference in salary [74].

An investigation of the impact of remittances on macro- and microeconomic growth for Romania and Moldova, using linear time regression models, showed that remittance-based economic growth is not sustainable and highlighted the long-term negative impact on the country of origin [5].

Using a multiple regression model in the analysis of remittance and GDP correlation for Romania and Bulgaria, the results indicated that there is no direct relationship between remittances/capita received and GDP/capita growth rate in Romania and Bulgaria [75]. The analysis of the impact of remittances sent by Romanian migrants on the country's economic growth for the period 2000–2016 using a simple linear regression model demonstrated the positive impact of remittances on GDP [18].

In this context, we chose to estimate the evolution of the economic indicators mentioned above by means of the following 3 polynomial-time regression models:

- the linear regression model:

$$y_t = a_0 + a_1 t + u_t. \quad t = 1, \ldots, n, \tag{1}$$

  which provides a general data trend;
- the quadratic regression model:

$$y_t = b_0 + b_1 t + b_2 t^2 + u_t. \quad t = 1, \ldots, n, \tag{2}$$

  providing a U-shape or inverted-U shape for the approximate curve,
- the cubic model:

$$y_t = c_0 + c_1 t + c_2 t^2 + c_3 t^3 + u_t. \quad t = 1, \ldots, n, \tag{3}$$

  determining a N-shape or inverted N shape approximate curve, according to the signs of the coefficients $t$, $t \hat{\ } 2$, and $t \hat{\ } 3$ and the relations between them.

The theoretical models were estimated, and the data were processed using the econometric, data processing, and analysis software.

The empirical results of the estimations obtained using the least square method in EViews 9.5 [73], are presented in Table 4.

**Table 4.** Estimation of parameters, period 2008–2017.

|  |  | S_REM_EU | S_REM_W | MG | REM_EU | REM_W | GDP | GDP_C |
|---|---|---|---|---|---|---|---|---|
| Model 1 | c | 2.4799 | 2.8230 | 239,611.5 | 3169.54 | 3590.233 | 116,632.3 | 5580 |
|  | t | −0.1691 | −0.1782 | −5524.66 | −166.733 | −166.295 | 5607.824 | 323.6364 |
|  | Trend | decreas. | decreas. | decreas. | decreas. | decreas. | increas. | increas. |
| Model 2 | c | 3.6792 | 3.8737 | 352329 | 5299.323 | 5570.067 | 146,233 | 7063.333 |
|  | t | −0.7687 | −0.7036 | −61,883.4 | −1231.62 | −1156.21 | −9192.5 | −418.03 |
|  | $t^2$ | 0.0545 | 0.0478 | 5123.52 | 96.80833 | 89.99242 | 1345.484 | 67.42424 |
|  | Shape | U | U | U | U | U | U | U |
|  | Turning point | 7.05 | 7.35 | 6.03 | 6.36 | 6.42 | 3.41 | 3.10 |
| Model 3 | c | 4.2764 | 4.4442 | 370,329.9 | 6410.227 | 6721.577 | 160,184.6 | 7693.333 |
|  | t | −1.2984 | −1.2096 | −77,849.3 | −2216.94 | −2177.54 | −21,566.8 | −976.807 |
|  | $t^2$ | 0.1694 | 0.1575 | 8585.24 | 310.4436 | 311.4367 | 4028.481 | 188.5781 |
|  | $t^3$ | −0.0069 | −0.0066 | −209.80 | −12.9476 | −13.4209 | −162.606 | −7.34266 |
|  | Shape | inv. N | inv. N | inv. N | inv. N | inv. N | inv. N | inv. N |
|  | FTP | 6.21 | 6.59 | 5.74 | 5.38 | 5.33 | 3.36 | 3.18 |
|  | STP | 10.00 | 9.19 | 21.54 | 10.60 | 10.13 | 13.15 | 13.94 |

Note: decreas.—decreasing; increas.—increasing; inv. N—inverted N.

The analysis of the linear model shows that 5 of the analyzed indicators (S_REM_EU, S_REM_W, MG, REM_EU, and REM_W) have a decreasing tendency, while GDP and GDP per capita have an increasing tendency. According to the quadratic model, the approximate curves for all the indicators are U-shaped, as the estimated parameters for the quadratic term are positive, having minimum values at the critical points located inside of the analyzed period. This means that, after a period of decrease in the first years of the analyzed period, all the indicators show an increasing trend.

For the cubic equation, there are two critical values of the variable, FTP (first turning point) and STP (second turning point), while the approximating curves have an inverted N-shape, as the estimated parameters of the third order terms are all negative. This means that the estimated variation shows a decrease to the first critical point, then an increase to the second critical point, followed by a decrease. Based on the estimation models, we can say that the upward trend is maintained for the period between 2008 + FTP and 2008 + STP, followed by a downward trend.

The cubic regression model estimates that migration should have an upward trend for an even longer period, until 2030 (2008 + 22). It also estimates that there is a growth period for GDP until 2021 (2008 + 13) and GDP_C until 2022 (2008 + 14), after which the trend will be decreasing. Regarding the share of remittances in GDP, it is noted that, for both remittances at the EU level (2008 + 9) and for remittances at that global level, they are already in the period of decline (after 2008 + 10). Let us point out that, as the data covers only 10 years, long time estimations are not reliable.

The accuracy of the estimation is illustrated by the values of R-squared indicator, presented in Table 5. As the values of this indicator are closed to 1 for models 2 and 3, we may conclude that these models are statistically significant.

**Table 5.** Values of R-squared for models 1–3.

| R2 | S_REM_EU | S_REM_W | MG | REM_EU | REM_W | GDP | GDP_C |
|---|---|---|---|---|---|---|---|
| Model 1 | 0.5596 | 0.6418 | 0.1493 | 0.2743 | 0.2951 | 0.6926 | 0.7511 |
| Model 2 | 0.9319 | 0.9346 | 0.9710 | 0.8663 | 0.8483 | 0.9478 | 0.9598 |
| Model 3 | 0.8151 | 0.8389 | 0.9791 | 0.9282 | 0.9202 | 0.9696 | 0.9743 |

### 3.3. Estimation of the Impact of Migration on Remittances

To investigate the influence of migration on the value of remittances at the EU level as well as globally, we used the following two linear difference equation models, concerning only the migration (MG) and remittances (REM_EU, REM_W):

$$Y_t = \alpha_1 Y_{t-1} + \alpha_2 X_t + u_t, \tag{4}$$

$$Y_t = \beta_1 Y_{t-1} + \beta_2 X_t + \beta_3 X_{t-1} + u_t, \tag{5}$$

where $Y_t$ represents the remittances at the European Union level or globally; $t = 1, \ldots, T$, refers to the time period; and $X_t$ represents the migration at the time $t$. The term $\alpha_1 Y_{t-1}$ represents the remittances from the emigrants prior to the reference year, with $\alpha_2 X_t$ representing the remittance contribution of the migrants from the current year.

In Equation (5), the term $\beta_1 Y_{t-1}$ represents the remittances from the emigrants prior to the reference year, $\beta_2 X_t$ represents the contribution of remittances from migrants from the current, while $\beta_3 X_{t-1}$ represents the contribution of remittances of migrants from the previous year.

The results of the empirical estimations obtained using EViews are given in Tables 6 and 7.

**Table 6.** Empirical estimation of parameters in Equation (4).

| Equation (4) (REM_EU) | | | | Equation (4) (REM_W) | | | |
|---|---|---|---|---|---|---|---|
| **Variable** | **Coefficient** | **Std. Error** | **Prob.** | **Variable** | **Coefficient** | **Std. Error** | **Prob.** |
| REM_EU(-1) | 0.108183 | 0.0572 | 0.100 | REM_W(-1) | 0.074744 | 0.0553 | 0.2186 |
| MG | 0.008715 | 0.0007 | 0 | MG | 0.011028 | 0.0008 | 0 |
| R-squared | 0.89222 | | | 0.879586 | | | |

**Table 7.** Empirical estimation of parameters in Equation (5).

| Equation (5) (REM_EU) | | | | Equation (5) (REM_W) | | | |
|---|---|---|---|---|---|---|---|
| **Variable** | **Coefficient** | **Std. Error** | **Prob.** | **Variable** | **Coefficient** | **Std. Error** | **Prob.** |
| REM_EU(-1) | 0.071372 | 0.1135 | 0.552 | REM_W(-1) | −0.08758 | 0.096 | 0.396 |
| MG | 0.007921 | 0.0022 | 0.011 | MG | 0.008335 | 0.002 | 0.002 |
| MG(-1) | 0.001174 | 0.0031 | 0.714 | MG(-1) | 0.004726 | 0.002 | 0.100 |
| R-squared | 0.894815 | | | 0.926084 | | | |

The values of parameters estimated for Equation (4) related to the remittances from EU could be interpreted as follows: the remittances in the current year consist of an amount of 10.8% from the remittances of the previous year coming from older emigrants, while the contribution of the new migrants (from the current year) is about 700 euros/month/capita (8700 euros/year/capita). The estimation of Equation (4) with data on global remittances shows similar results, the monthly amount being about 900 euros/month/capita (11,000 euros/year).

The results of the estimation in Table 7, concerning remittances from EU, could have the following interpretation: the contribution of migrants from this year is about 650 euros/month/capita (7900 euros/year/capita), and the contribution of emigrants from the previous year decreases to about 100 euros/month/capita (1200 euros/year/capita), while the contribution of older migrants decreases to an amount of 7.14% from the remittances of the previous year. For the global remittances, the impact of the term $\beta_1$ is negative, while the contribution of migrants in the current year is estimated at about 700 euros/month/capita and that of the previous year is at 400 euros/month/capita.

According to the results of the estimation, the amount of individual remittances decreases as the period of stay is longer.

The cointegration of variables in the two equations is justified by the results of the ADF (Augmented Dickey–Fuller) unit root test on the stationarity of the residuals, presented in Table 8.

**Table 8.** ADF test results for the residual values of the estimations of Equations (4) and (5).

| Variable | Equation (4) | | Equation (5) |
|---|---|---|---|
| | Level | First Difference | Level |
| | Prob | Prob | Prob |
| REM_EU | 0.048 | | 0.016 |
| REM_W | | 0.086 | 0.004 |

According to the data in Table 8, the residuals series are stationary, which means that the cointegration hypothesis on the equations is verified.

## 4. Conclusions

In line with the objectives of the Europe 2020 strategy, the employment strategy must create safer and better jobs throughout the European Union, given that the working population in the EU28 has been steadily declining. Although after 1990, the number of migrants in the developed countries of the European Union has increased considerably, it will continue to grow as long as there is a demand for labor and stronger economic development in the developed countries than in the countries from where they come. As for Romania, the causes of migration were mostly economic in nature (the concrete needs of the household and the economic problems at the Romanian level) and the need to try something new by imitating the success models of other migrants in the community. In recent years, there is a change in the destination countries of Romanian migrants from Italy, Spain, and Germany, to England, Norway, Finland, Sweden, and Denmark.

Analyzing the origin of remittances, it is observed that the vast majority (over 80%) come from EU countries, which means that Romanian citizens migrate to find a job, especially to developed EU countries.

The amount of remittances entered in Romania both from the global level and from the level of the EU states was in a permanent increase until the global crisis started, when the value of the remittances decreased dramatically by over 40%, a decline also recorded in other EU emerging countries [76], followed in recent years by a continuous growth, which is confirmed by the European Commission Country Report [7].

The evolution of GDP and GDP per capita shows a permanent increase in the analyzed period, meaning that, in Romania, there has been an improvement in the standard of living while the remittances highly diminished to almost half. Although GDP and GDP per capita have seen a permanent increase, except for 2009, in terms of the share of remittances in GDP, a sharp decrease is observed in 2017 compared to 2008 (about 60%) due to the financial crisis that has determined on the one hand a reduction in the number of departures and because the citizens sent less money to the country.

Estimations of the evolution of the indicators using linear, quadratic, and cubic regression models showed that the quadratic and cubic models provide similar results. Thus, by the quadratic model, all the macroeconomic indicators followed a U-shaped form evolution, with the critical minimum point located inside the analyzed period. The cubic model showed an inverted N-shaped evolution with maximum point located at the end or outside of the period. Consequently, the estimations results show that three of the indicators have an upward trend, namely migration until 2030 and GDP and GDP_C by 2022, after which the trend will be decreasing, while the share of remittances in GDP are in the phase of decline (S_REM_EU and S_REM_W from 2018).

The estimation of the impact of migration on remittances using two linear difference equation models lead to the conclusion that the value of the individual remittances sent home, in Romania, from both EU and world level have a tendency to decrease after the first year. Thus, if in the first year each citizen sends around 650 euros per month (those in the EU) or 700 euros/month globally, in the following years, the value of remittances from the EU country decreases to 100 euros/month/capita and respectively 400 euros per global. The decrease could be motivated either by the fact that migrants settle

permanently in the country of residence or because their financial surpluses diminish as a consequence of change or loss of the job.

Thus, labor force migration in developed countries both in the European Union and worldwide has produced and still produces a very complex set of economic effects at the level of Romania. The emigration of some members of the households who are sending remittances has determined an increase of the income of the remaining members in the country. These supplementary incomes could be used for consumption and for various investments and access to services, health, and education, thus leading to an economic development of the respective community. Remittances represented and still represent a relatively stable source for Romania, although their value has a decreasing tendency (with a drastic decrease during the financial crisis), affecting in a positive way the standard of living of Romanian citizens. In this sense, remittances entered in Romania can have positive effects on the increase of the standard of living of those receiving them and on the local economic development (by increasing consumption and investments).

Among the negative effects of remittances and possible risks for Romania could be mentioned the increase of inequalities at the community level, the decrease of the intention of employment, the dependence on the amounts of money received from emigrants, the appearance of inflationary pressure (the excessive demand for land and houses leads to artificial growth of prices), and the exodus of "brains" and skilled workers.

This study referred to the effects of remittances on the economic development of Romania, but these aspects can be extended to the other states of central and southeastern Europe. In developing countries, such as Romania, the labor market is less attractive, which, together with the possibility of external mobility for work, has led to profound numerical and structural imbalances. This fact may jeopardize the possibility of future generations to achieve sustainable long-term economic growth.

The ever-increasing levels of remittances of migrants to the countries of origin have made them one of the main engines for promoting the globalization and economic growth of the developing countries. In this sense, we can say that migration and remittances can act as mechanisms for adjusting the flows of labor resources between the countries of origin and those of destination. On the one hand, they are a consequence of the failure of the national policy of the country of origin manifested by the inability to meet the individual needs regarding the decent employment opportunities and the income obtained [74], and on the other, they can be a tool to support the economic growth. The impact of remittances for the countries of origin is significant, at least from the perspective of the beneficiary households. By transferring large amounts of money, information, ideas, and practices home, migrants can make significant changes in their home countries and communities.

Romania's policy, as well as the policy of Eastern European countries, should consider restructuring and strengthening economic institutions and policies in order to create an environment that encourages the labor force not to leave the country, to stimulate migrants to return, and even to attract skilled labor from other countries. At the same time, government policy needs to take steps to make better use of remittances, which could be used predominantly for investment and less for consumption; to improve the use of the remaining workforce; and to address the fiscal implications of emigration.

According to Eurostat statistics and forecasts, in the future, the flow of migrants from Eastern Europe to the developed EU countries will continue; thus, in order to compensate for the negative effect of emigration on the countries of origin, appropriate policies will have to be adopted at the EU level regarding the transfer mechanisms within the policy of EU structural and cohesion funds that will reduce regional disparities and accelerate convergence.

**Author Contributions:** All three authors equally contributed in designing and writing this paper. All authors have read and agreed to the published version of the manuscript.

**Funding:** This research received no external funding.

**Acknowledgments:** The authors express their gratitude to the anonymous reviewers whose thoughtful comments led to improvement of this paper. This research did not receive any specific grant from funding agencies in the public, commercial, or not-for-profit sectors.

**Conflicts of Interest:** The authors declare no conflict of interest.

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
