# Peer review of "Remittances, Migration and Gross Domestic Product from Romania’s Perspective"

_sustainability, doi:10.3390/su12010212_

Round 1

Reviewer 1 Report

An interesting, well-written study. But language use must certainly be improved, though it is not my job to judge it.

It would be worthwhile to introduce East Central Europe in more detail. How special is the Romanian route? To what extent can the experience be generalized to other groups of countries?

Author Response

To the Editor of Sustainability, from the Authors of manuscript Sustainability-669973

December 16, 2019

The authors are grateful to the Academic Editor and reviewers of manuscript number Sustainability-669973 for their thoughtful comments.

All of the comments of the reviewers are taken into consideration and answered, point by point, in this letter. The authors have introduced a labelling into the comments for reference purposes.

The authors believe that the revised version of the paper is better than the originally submitted one. For this reason, they reiterate their gratitude to the anonymous reviewers.

In paper revision we used the "Track Changes" function in Microsoft Word.

Response to Reviewer 1 Comments

General Comments: “An interesting, well-written study.”

Authors: The authors are greatly encouraged by the nice opinion concerning the idea of the article. ” But language use must certainly be improved, though it is not my job to judge it.” Authors: We have re-read the whole paper and carefully corrected the English. We hope that the resulting version improves the initial one. “It would be worthwhile to introduce East Central Europe in more detail.” Authors: We hope that the addition to introduction (lines 37-55) and conclusion (lines 594-598) answer to this suggestion. “How special is the Romanian route?” Authors: We hope that the addition to introduction (lines 74-83, 89-96) answer to this suggestion. “To what extent can the experience be generalized to other groups of countries? “ Authors: We reformulate some conclusions and added corresponding comments in the last section (lines 550-554, 594-609).

Concluding Comments by the authors

The authors have considered and commented all the queries raised by the reviewer of their manuscript number Sustainability-669973. Improvements are presented in the new version of the paper. The authors believe that the present version of the paper is better than that originally submitted. For this reason, they reiterate their gratitude to the Reviewer.

Reviewer 2 Report

The paper is interesting and the introduction provides a good motivation for the empirical analysis conducted in the empirical part of the paper.

The methods used are adequate to the objective and allow to answer the main question under study.

My general opinion about this paper is positive. Neverthless, I believe that some key aspects could be improved:

(i) the final section of the paper could discuss in more detail the most important policy implications emerging from the study;

(ii) the authors could provide a stronger link between the evidence obtained and previous results on the same topic;

(iii) please explain better the importance of the topic for the specific case of Romania;

(iv) I would like to see some discussion about the potential generalization of the results obtained by the authors in this study.

Author Response

To the Editor of Sustainability, from the Authors of manuscript Sustainability-669973

December 16, 2019

The authors are grateful to the Academic Editor and reviewers of manuscript number Sustainability-669973 for their thoughtful comments.

All of the comments of the reviewers are taken into consideration and answered, point by point, in this letter. The authors have introduced a labelling into the comments for reference purposes.

The authors believe that the revised version of the paper is better than the originally submitted one. For this reason, they reiterate their gratitude to the anonymous reviewers.

In paper revision we used the "Track Changes" function in Microsoft Word.

Response to Reviewer 2 Comments

General Comments: The paper is interesting and the introduction provides a good motivation for the empirical analysis conducted in the empirical part of the paper. The methods used are adequate to the objective and allow to answer the main question under study.

Authors: The authors are greatly encouraged by the nice opinion concerning the idea of the article.

My general opinion about this paper is positive. Nevertheless, I believe that some key aspects could be improved:

“The final section of the paper could discuss in more detail the most important policy implications emerging from the study.“ Authors: We reformulate some conclusions and added corresponding comments in the last section (lines 599-621). “The authors could provide a stronger link between the evidence obtained and previous results on the same topic.“ Authors: The section “Literature review” was extended (lines 200-210, 217-234, 245-254). At the same time, section 3.2 “Estimation of evolution by means of polynomial-time regression” was reorganized and lines 438-463, referring to models, were added. Also, some connections between the obtain results and previous works were emphasized (line 550-554). “Please explain better the importance of the topic for the specific case of Romania.“ Authors: We hope that the addition to introduction (lines 74-83, 89-96, 107-109) answer to this suggestion.

  “I would like to see some discussion about the potential generalization of the results obtained by the authors in this study.“

Authors: We hope that the addition to introduction (lines 130-136, 139) and to conclusion (lines 550-554, 594-609) answer to this suggestion.

Concluding Comments by the authors

The authors have considered and commented all the queries raised by the reviewer of their manuscript number Sustainability-669973. Improvements are presented in the new version of the paper. The authors believe that the present version of the paper is better than that originally submitted. For this reason, they reiterate their gratitude to the Reviewer.

Reviewer 3 Report

Recently there is revived interest in the problem of remittances and their linkages with macroeconomic fundamentals. The present paper builds in this direction, focusing on the case of Romania. Despite an interesting topic, the manuscript suffers from a series of flaws, both major and minor, on which I will comment below. 

Major problems: 

The title of the manuscript requires upgrades as it is not correct in the present version.  Upgrades are required also in the abstract which is poorly constructed and a little far from the specifics of such a section. The introduction is one of the poorest sections of the manuscript. It states little and manages to detect only a limited gap in the literature. In addition to this, I would like to see a clear formulation of the original contributions of the manuscript which at this point appear to be rather thin. This leads to my main concern on the manuscript which is the fact that I can't detect a clear research question that the authors are trying to put forward. The literature review section is constructed more as a background section, addressing little actual literature dealing with modeling the impact of remittances. I don't have anything against this approach but, if the purpose is to generate a literature review, heavy upgrades are needed (especially given the numerous investigations tackling exactly this topic). Several areas could be removed, as their message can be considered as rather trivial by a certain public. Statements such as those in lines 155 - 158 can be considered as general knowledge if not trivial comments.  Section 3 is very poor in terms of both approach and content. This is my second major problem with this manuscript as the method employed can be regarded as casual at best. I can't detect traces of originality or added value in this approach. I will not go into additional details and critiques about the length of the sample employed. The results are in close correspondence with the modeling approach. Section 4.1 can be removed without losing substance as its intrinsic value is extremely limited. The remainder of the section reports the results obtained under the specifications already mention. I will not insist on its general relevance.  The Conclusions section is a bit better than the precedent ones. Still, I would normally like to see some clear take-aways on the topic. 

Minor problems:

 I advise the authors to have the manuscript revised by a native speaker before resubmitting. The present shape of the manuscript demonstrates that they do not have a proper command of English grammar and academic writing.   I do not understand the logic of lines 88 - 90 after the paragraph dealing with the structure of the manuscript. I can't grasp the logic behind reference number 42 hinting to the use of Eviews. 

Author Response

To the Editor of Sustainability, from the Authors of manuscript Sustainability-669973

December 16, 2019

The authors are grateful to the Academic Editor and reviewers of manuscript number Sustainability-669973 for their thoughtful comments.

All of the comments of the reviewers are taken into consideration and answered, point by point, in this letter. The authors have introduced a labelling into the comments for reference purposes.

The authors believe that the revised version of the paper is better than the originally submitted one. For this reason, they reiterate their gratitude to the anonymous reviewers.

In paper revision we used the "Track Changes" function in Microsoft Word.

Response to Reviewer 3 Comments

General Comments: Recently there is revived interest in the problem of remittances and their linkages with macroeconomic fundamentals. The present paper builds in this direction, focusing on the case of Romania.

Authors: The authors are greatly encouraged by the nice opinion concerning the idea of the article.

Despite an interesting topic, the manuscript suffers from a series of flaws, both major and minor, on which I will comment below. 

Major problems: 

“The title of the manuscript requires upgrades as it is not correct in the present version. “ Authors: The authors agree with this suggestion, we believe that the new title “Remittances, migration and gross domestic product from Romania’s perspective” reflects better the content of the paper. “Upgrades are required also in the abstract which is poorly constructed and a little far from the specifics of such a section.“ Authors: The authors agree with this recommendation, the abstract was upgraded. “The introduction is one of the poorest sections of the manuscript. It states little and manages to detect only a limited gap in the literature. In addition to this, I would like to see a clear formulation of the original contributions of the manuscript which at this point appear to be rather thin. This leads to my main concern on the manuscript which is the fact that I can't detect a clear research question that the authors are trying to put forward.“ Authors: To answer these suggestions the introduction was restructured: references to literature related to the topics were added (lines 37-51, 80-96), the structure and content of the study, as well as the original contribution were reformulated (lines 107-139). “The literature review section is constructed more as a background section, addressing little actual literature dealing with modelling the impact of remittances. I don't have anything against this approach but, if the purpose is to generate a literature review, heavy upgrades are needed (especially given the numerous investigations tackling exactly this topic). Several areas could be removed, as their message can be considered as rather trivial by a certain public.“ Authors: We completed the References section with 34 new titles, most of them referred in the section “Literature review”, which was extended (lines 200-210, 217-234, 245-254). “Statements such as those in lines 155 - 158 can be considered as general knowledge if not trivial comments.“ Authors: The authors agree with this opinion, those lines were eliminated. “Section 3 is very poor in terms of both approach and content. This is my second major problem with this manuscript as the method employed can be regarded as casual at best. I can't detect traces of originality or added value in this approach. I will not go into additional details and critiques about the length of the sample employed. The results are in close correspondence with the modeling approach. Section 4.1 can be removed without losing substance as its intrinsic value is extremely limited. The remainder of the section reports the results obtained under the specifications already mention. I will not insist on its general relevance.“ Authors: The authors are grateful for this recommendation. Section 3 and 4 were restructured into one section and completed with other relevant comments and references. We believe that this structure presents a better exposition of the analytical study and hope that it answers to the suggestions of the reviewer. “The Conclusions section is a bit better than the precedent ones. Still, I would normally like to see some clear take-aways on the topic.“ Authors: The author are encouraged by this nice opinion. The Conclusion section was extended by lines 594-621.

Minor problems:

“I advise the authors to have the manuscript revised by a native speaker before resubmitting. The present shape of the manuscript demonstrates that they do not have a proper command of English grammar and academic writing.“ Authors: The paper was revised and carefully corrected. We believe that the resulting version improves the initial one. “I do not understand the logic of lines 88 - 90 after the paragraph dealing with the structure of the manuscript.“ Authors: We responded to this suggestion by reformulating the structure and content of the study (lines 107-139). “I can't grasp the logic behind reference number 42 hinting to the use of Eviews.“ Authors: The reference to the software EViews was clarified in lines 377-378.

Concluding Comments by the authors

The authors have considered and commented all the queries raised by the reviewer of their manuscript number Sustainability-669973. Improvements are presented in the new version of the paper. The authors believe that the present version of the paper is better than that originally submitted. For this reason, they reiterate their gratitude to the Reviewer.

Round 2

Reviewer 2 Report

The new version of the papers answers my previous concerns.

Therefore, I think the paper can be accepted for publication.

Reviewer 3 Report

First of all, I would like to congratulate the authors for their efforts in trying to produce a better version of the manuscript. My comments on the revision will follow the observations put forward in my initial review.

Major problems:

Comment 1: In the current version, the title is grammatically correct. Despite this fact, I do not know if the current version exhibits the best choice of words for the title of academic work. It can be argued simply that it makes little sense, but I will not go into such details.

Comment 2: I acknowledge the current version of the abstract as passable, although I would have expected more attention to be given to this matter.

Comment 3: The authors either fail to completely understand my initial observation or ignored what I highlighted as my main concern in the manuscript.

I see little efforts put into highlighting the original component of the article. Moreover, I still fail to detect a clear-cut and relevant research question. Coming back to the introductory section, I can't say that the alterations conducted can be considered as upgrades, given the fact that the section is in the same position as the original.

Comment 4: I consider the upgrades in this section to be passable.

Comment 5: Again, I consider that the authors either ignore or do not grasp the meaning of the previous comments. In comment number five I was stating as my second major concern the fact that the entire method is lax at best. Again, I do not detect any efforts in providing a more suitable version. I am highlighting the fact that from a technical point of view the revised version is in the same place as the original version.

Comment 6: I acknowledge efforts in restructuring the section. However, my main observations on content remain.

Comment 7: The alterations in the Conclusions section can be regarded as passable.

Minor problems:

Comment 1: As stated earlier the manuscript needs to be reviewed by a native speaker or by a person with a decent command of English. The vast majority of language problems remain.

Comment 2 and Comment 3 can be considered as solved.